# Evaluation of Antibacterial and Antifungal Properties of Low Molecular Weight Chitosan Extracted from *Hermetia illucens* Relative to Crab Chitosan

**DOI:** 10.3390/molecules27020577

**Published:** 2022-01-17

**Authors:** Adelya Khayrova, Sergey Lopatin, Balzhima Shagdarova, Olga Sinitsyna, Arkady Sinitsyn, Valery Varlamov

**Affiliations:** 1Institute of Bioengineering, Research Centre of Biotechnology, Russian Academy of Sciences, 119071 Moscow, Russia; lopatin@biengi.ac.ru (S.L.); shagdarova.bal@gmail.com (B.S.); varlamov@biengi.ac.ru (V.V.); 2Entoprotech Ltd., Skolkovo Innovation Centre, 121205 Moscow, Russia; 3Department of Chemistry, Moscow State University, 119991 Moscow, Russia; oasinitsyna@gmail.com (O.S.); apsinitsyn@gmail.com (A.S.)

**Keywords:** chitosan, insect, depolymerisation, antibacterial activity, antifungal activity, *Hermetia illucens*

## Abstract

This study shows the research on the depolymerisation of insect and crab chitosans using novel enzymes. Enzyme preparations containing recombinant chitinase Chi 418 from *Trichoderma harzianum*, chitinase Chi 403, and chitosanase Chi 402 from *Myceliophthora thermophila*, all belonging to the family GH18 of glycosyl hydrolases, were used to depolymerise a biopolymer, resulting in a range of chitosans with average molecular weights (M_w_) of 6–21 kDa. The depolymerised chitosans obtained from crustaceans and insects were studied, and their antibacterial and antifungal properties were evaluated. The results proved the significance of the chitosan’s origin, showing the potential of *Hermetia illucens* as a new source of low molecular weight chitosan with an improved biological activity.

## 1. Introduction

Chitin is one of the most abundant polysaccharides on Earth. Chitosan, known as chitin’s deacetylated derivative, has attracted much research attention due to its extensive applications in ecology, agriculture, food industry, cosmetic, and biomedical fields [1,2,3,4,5]. Over the last couple of decades, the main sources of commercial chitin and chitosan have been crustaceans; however, alternative sources such as insects have recently been emerging.

Black soldier fly or *Hermetia illucens* larvae have been applied as a promising tool for waste treatment with the production of feed protein, fat, and organic fertiliser [6,7,8]. In addition, this technology is environmentally more efficient compared to other waste disposal methods, e.g., resulting in 47 times lower direct CO_2_-eq emissions than for composting [9]. Thus, insect chitin, a byproduct for the feed industry, is currently investigated as a potential source of new biopolymer properties and applications [10].

It has been discussed in the literature that decreasing chitosan’s molecular weight helps to improve its solubility and biological activity [11,12]. Several methods have been explored to depolymerise chitosan, including chemical and physical depolymerisation processes [13,14]. The chemical degradation of chitosan is usually carried out at elevated temperatures using strong acids or alkali, and, therefore, can cause equipment corrosion, tedious product separation, and the treatment of acid wastewater [15]. Physical methods applying sonic radiation or hydrodynamic shearing can lead to uncontrolled degradation of chitosan-generating products. Enzymatic hydrolysis, on the contrary, is an environmentally friendly method that enables the controlled hydrolysis of chitosan performed by naturally designed proteins for practical purposes [16].

This study reveals the results on commercial crab chitosan and chitosan obtained from *Hermetia illucens* larvae and their molecular weights, controlled by the enzymatic hydrolysis method. Chitosan samples were physicochemically characterised, and their purity was confirmed. The antibacterial and antifungal properties of insect chitosan were also examined and compared to the crab chitosan.

## 2. Results

### 2.1. Obtaining Chitin and Chitosan

The scheme for obtaining chitin from the larvae of *H. illucens* is standard and includes demineralisation and deproteinisation steps. The demineralised biomass was sieved and divided into different fractions—more than 5 mm, 2–5 mm, and less than 2 mm. Each fraction was separately deproteinised, and the amounts of chitin were determined: the values were 30, 14, and 2%, respectively. This indicates a negligible content of chitin-containing residue in the smallest fraction, which was not subjected to further processing due to economic inexpediency but could further be used as a feed additive.

Chitin was subsequently deacetylated to chitosan. After carrying out demineralisation, deproteination, and deacetylation steps, some of the impurities were still present in the final product. The resulting material contained approximately 4% of the fraction insoluble in acetic acid. Presumably, this indicates the presence of “cuticulin”, a substance occurring in the exocuticle and epicuticle (Figure 1) [17]. Cuticulin contains a fatty or waxy component and is difficult to hydrolyse during the deacetylation reaction [18]. According to Wigglesworth [19], the epicuticle contains a lipid material chemically incorporated in its substance, probably as a lipoprotein.

The chitosan obtained in this way requires additional purification by reprecipitation from a solution in acetic acid. The resulting chitosan was characterised by HPLC. The weight-average molecular weight (M_w_) was 480–570 kDa and the polydispersity index (PDI) was 1.8–2.1. Degree of deacetylation (DDA) of chitosan was calculated as 87–92% using the method of conductometric titration. With repeated deacetylation, the content of cuticulin was reduced to 0.64% and DDA increased to 97%; however, M_w_ decreased to 350 kDa. Therefore, it can be concluded that the removal of insoluble impurities by increasing the processing time was not reasonable to obtain high molecular weight chitosan.

### 2.2. Enzymatic Depolymerisation of Chitosan

The purpose of this experiment was to study the products of hydrolysis of chitosan isolated from the larvae of *H. illucens* in comparison with the hydrolysates of crab chitosan using three new enzyme preparations with chitinolytic activity due to the increased content of recombinant *Myceliophthora thermophila* chitosanase, *Trichoderma harzianum,* and *M. thermophila* chitinases.

Figure 2 and Figure 3 show the dynamics of enzymatic hydrolysis of chitosans obtained from *H. illucens* larvae and crab, respectively. Based on the kinetic curves of the accumulation of the conversion products of chitosans, the hydrolysis time to obtain chitosans with specified molecular weights was estimated.

Using enzymatic hydrolysis, low molecular weight chitosans of a given composition (6–21 kDa) can be obtained and further used for studying their biological activity. The purity of the obtained chitosan samples was confirmed by elemental (Table 1) and ^1^H NMR (Figure 4) analyses.

### 2.3. Antibacterial and Antifungal Activities

The antibacterial activities of depolymerised crab and insect chitosans were studied against Gram-positive *S. epidermidis* and Gram-negative *E. coli*.

The results of the experiment with *S. epidermidis* are presented in Table 2. The studied chitosan samples had a low polydispersity index (PDI = 1.5–2). Differences in PDI values of the studied chitosan pairs did not exceed 10–15%, corresponding to the correctness in the comparison of samples. The minimum inhibitory concentration (MIC) values for the studied samples were in the range of 62.5–125 μg/mL. For the low molecular weight chitosan obtained from *H. illucens* larvae, the MIC value was 62.5 μg/mL. In the case of *E. coli,* all chitosans showed a weaker antibacterial activity with MIC values exceeding 500 μg/mL.

The antifungal activity was evaluated by determining the effect of the chitosans on the metabolic activity of two phytopathogens—*B. cinerea* and *F. oxysporum*. The antifungal activity of the chitosan samples was studied in the concentration range of 0.11–1.8 μg/mL.

The metabolic activity (MA) of *F. oxysporum* is shown in Figure 5. The MA was evaluated based on the concept of IC_50_, the concentration of half-maximal inhibition causing 50% inhibition of microorganism growth (these areas are highlighted in red) [20].

The metabolic activity of *B. cinerea* was greater than or equal to 50% at the studied chitosan concentrations.

## 3. Discussion

Due to the harsh process parameters (pH and temperature), chemical hydrolysis can lead to changes in the product and can have a negative impact on the environment [21,22]. In this regard, enzymatic hydrolysis represents a potential alternative, providing milder process conditions and reducing waste generation [23]. In nature, chitin and chitosan can be depolymerised by specific enzymes such as chitinases, chitosanases, and glucoaminidases [24]. According to the enzyme nomenclature committee, chitinases (EC 3.2.1.14) are defined as enzymes capable of performing endohydrolysis of b-1,4-linkages in chitin [11]. Chitosanases (EC 3.2.1.132) constitute a family of enzymes capable of performing endohydrolysis of a b-1,4-glycosydic bond between glucosamine residues in partially acetylated chitosan, from the reducing end [11]. These proteins have currently been applied for biotechnological transformation towards low molecular weight chitosans and chitooligosaccharides with desired physicochemical and biological properties [25,26].

The results of the enzymatic hydrolysis revealed that upon the treatment of chitosans from the crab and the black soldier fly with the enzyme preparations of chitosanase and chitinase from *M. thermophila*, the obtained chitosan oligomers have comparable M_w_. Nevertheless, the hydrolysates of the chitosans obtained under the same conditions using chitinase *T. harzianum* differ in M_w_ by almost 1.5 times (e.g., 14,995 Da versus 9959 Da). This confirms that chitosans from different sources differ in their properties; therefore, the development of methods for producing chitosan from insects is of high importance. The results of the analysis of the molecular weight characteristics of chitosan oligomers show that in all cases, the accumulation curves of the reaction product reach a plateau, and further thermostating does not lead to a significant increase in the degree of conversion of the substrate after 24 h of incubation provided the same ratios of chitosan to enzyme (5 mg of protein per 1 g of substrate).

It was earlier reported that chitosan and its derivatives possess antimicrobial properties against fungi [27,28,29,30,31], bacteria [32,33,34], and viruses [35,36,37,38]. However, the mechanism of their antimicrobial activity has not yet been fully explained. According to Xing et al. [39], the antimicrobial mechanisms of chitosan involve electrostatic interactions, plasma membrane damage, interaction with DNA/RNA, metal chelation, and deposition onto the microbial surface.

The antibacterial activity of chitosan can be caused by its ionic interaction on the surface and inside the bacterial cells [40]. The positive charges of chitosan-based materials ionise the negatively charged molecules on the bacterial cells’ surface [40]. Furthermore, the higher activity of chitosan degradation products in relation to the high molecular weight biopolymer is explained by the possibility of the low molecular weight chitosan to penetrate inside the bacterial cells and to inhibit the protein synthesis due to its ability to interact with the negatively charged mRNA and block its further processing inside the cell [41,42,43]. However, most studies mentioned earlier have investigated chitosan obtained from crustaceans. This study compares the antimicrobial properties of depolymerised chitosan obtained from two sources—crustaceans and insects.

It was shown that the obtained chitosans present different antibacterial activities against different bacterial strains. All chitosans showed a weak antibacterial activity against *E. coli* (MIC was more than 500 μg/mL), while stronger antibacterial properties were demonstrated against *S. epidermidis*. This could be explained by the contrasting mechanisms of the interaction of chitosan with Gram-positive and Gram-negative bacterial cells due to their structural differences, as well as their electrostatic effects [44]. Gram-positive bacteria possess a cytoplasmic membrane covered with a cell wall formed of many layers of peptidoglycan, and the oppositely charged depolymerised chitosan can be bound to it. This causes the deformation of the bacterial cell wall, which, in turn, is associated with the exposure of the cytoplasmic membrane to osmotic shock, the burst of the cytoplasm, and ultimately the death of bacteria [45]. Contrary to Gram-positive bacteria, the Gram-negative bacterial cell contains an additional outer membrane consisting of lipopolysaccharides and proteins, which may be hard for the chitosan to penetrate.

The antifungal activity of chitosan is beneficial for the control of crop diseases and the increase of crop yields [46]. *B. cinerea* infects over 200 plant species by causing grey mould, which leads to significant financial losses (USD 10–100 billion annually) [47]. *F. oxysporum* causes vascular wilt diseases on herbaceous and woody plants, especially in warm temperate and tropical regions [48].

Chitosans obtained from *H. illucens* larvae show a metabolic activity of *F. oxysporum* <50% at a lower concentration than the chitosans from crab shells. Chitosan samples from *H. illucens* larvae, at concentrations from 0.23 mg/mL and M_w_ of 19.64 kDa and 21.36 kDa, exhibit MA of 12 and 15%, respectively, while chitosan samples from crab shells, at the same concentrations, have MA of 50% or more. Crab chitosans at concentrations of 0.45 mg/mL and M_w_ of 10.66 kDa and 16.15 kDa exhibit MA equal to 17 and 16%, respectively. Overall, chitosans extracted from *H. illucens* larvae showed a higher antifungal activity against *F. oxysporum*.

## 4. Materials and Methods

### 4.1. Materials

*Hermetia illucens* larvae of the fifth instar were provided by Entoprotech Ltd., Russia, Moscow. Then, 3000 g of larvae was blanched. The processed larvae went through oilpress (RawMID, Russia) treatment to remove the majority of lipids, proteins, and moisture. Then, the resultant larvae were lyophilised and kept in an air-tight plastic seal. The product contained approximately 6% lipids and 7% ash. Finally, 250 g of product was obtained (yield = 8%).

### 4.2. Chitin Extraction

Demineralisation: 1000 mL of 1% HCl was added to 100 g of the obtained material and stirred at 20 °C for 2 h. The solid residue was separated through a glass filter. It was washed with distilled water to neutral pH and lyophilised afterwards. Then, 49.2 g of product was obtained (yield = 49%). The dried demineralised matter was sieved (pore diameter = 2 mm) and weighed. The product yield was equal to 45%.

Deproteinisation and defatting: 320 mL of 30% (*w/w*) NaOH was added to 24 g of demineralised biomass, left at room temperature for 30 min, and then transferred to a 100 °C water bath and kept for 2 h with occasional stirring. Chitin was separated through a glass filter, washed with distilled water until neutral pH, and lyophilised. Finally, 4.8 g of product was obtained (yield = 20%).

### 4.3. Chitosan Preparation

Deacetylation: the deacetylation step was performed by using 320 mL of 50% (*w/w*) NaOH. The alkaline solution was added to 8.0 g of chitin and left at room temperature for 30 min. The suspension was warmed in a 100 °C water bath for 2 h with occasional stirring. The suspension was cooled, washed until neutral pH, and lyophilised. Finally, 6.5 g of chitosan was obtained (yield = 81%).

Purification: 6.5 g of chitosan was dissolved in 650 mL of 1% CH_3_COOH. The solution was filtered through a glass filter. Then, 1 M NaOH was added until pH 10 was achieved. The solution was dialysed in Spectra/Por Dialysis Tubular Membrane MWCO:10,000 (Spectrum Laboratories Inc., USA) and lyophilised. Then, 6.2 g of chitosan was obtained (yield = 95%). M_w_ = 500 kDa, DDA = 90%. The precipitate was collected, dried, and weighed. Finally, 0.26 g of precipitate was obtained (yield = 4%).

Crab (Heppe Medical Chitosan GmbH, Germany) chitosan was used as a comparison. M_w_ = 505 kDa, DDA = 66%.

### 4.4. Enzyme Preparations for Chitosan Depolymerisation

Complexes of extracellular enzymes containing chitinase Chi 418 from *Trichoderma harzianum*, chitinase Chi 403, and chitosanase Chi 402 from *Myceliophthora thermophila*, all belonging to the family GH18 of glycosyl hydrolases, were obtained on the basis of the fungus *Penicillium verruculosum* recombinant strains. These highly active producer strains (with secretory capacity up to 50–60 g/L of extracellular protein) were created according to the method described earlier [49], using the recipient strain of *Penicillium verruculosum* 537 (ΔniaD) in which the repression of the glucose catabolism was reduced [50,51,52]. The content of heterologous enzymes in the studied enzyme preparations was 40–50% of the total protein content in the culture liquid. The activity of the enzyme preparations towards chitosan (1000 kDa) is shown in Table 3.

### 4.5. Enzymatic Depolymerisation of Chitosans

The depolymerisation of crab chitosan (M_w_ 505 kDa, DDA 66%) and black soldier fly chitosan (M_w_ 502 kDa, DDA 88%) was carried out using three enzymatic preparations: #3-432.5 containing chitosanase Chi 402 from *M. thermophila*; #3-458.1 containing chitinase Chi 418 from *T. harzianum*; and #3-544.H containing chitinase Chi 403 from *M. thermophila*.

Chitosan samples were dissolved in a 0.1 M sodium acetate buffer (pH 5.0) to reach a final concentration of 10 g/L. The hydrolysis was carried out with constant stirring at 250 rpm on an Environmental Shaker Incubator ES-20 (BioSan, Riga, Latvia).

The initial volume of the reaction mixture was 30 mL. The reaction mixture contained chitosan (10 g/L) and an enzyme preparation (5 mg of protein per 1 g of substrate). A blank experiment was performed without the addition of the enzyme. The reaction mixture was incubated at 40 °C for 1, 2, 6, 24, and 48 h. The reaction was stopped by boiling for 10 min to inactivate the enzyme.

Purification of chitosan solutions after hydrolysis was carried out using a SEP-PAK C_18_ cartridge for rapid sample preparation (Millipore, Kenilworth, NJ, United States). After incubation, the solution was passed through the cartridge and washed with 50 mL of water. Then, the passed solutions and the wash were combined. The cartridge was regenerated by washing with 20 mL of 60% acetonitrile and 20 mL of water.

Then, 5% (*w/w*) NaOH was added to pH 9.5–10, the solution was dialysed against water in Spectra/Por Membrane MWCO:1000 (Spectrum Laboratories Inc., Rancho Domniguez, CA, USA), and lyophilised. The decrease of molecular weight was analysed by the HPLC technique.

### 4.6. Characterisation of Chitosan

#### 4.6.1. Elemental Analysis

The nitrogen, carbon, and hydrogen contents of insect and crab chitosans were determined with a CHNS Carlo Erba EA 1108 Analyser (Isomass, Calgary, AB, Canada).

#### 4.6.2. ^1^H NMR Analysis

^1^H NMR spectra were performed on a Bruker avance 700 MHz NMR spectrometer equipped with a triple resonance 5 mm PATXI 1H-13C/15N/D Z-GRD probe (Bruker, Billerica, MA, USA). The samples were dissolved in D_2_O, and the NMR spectra were recorded at 30 °C. The free induction decay acquisition time was 2.9 s, the number of time points was 65,536, the spectrum sweep width was 15.9 ppm, and the number of scans was8. The residual HOD water signal was not suppressed to avoid distortion of the integral values. ^1^H chemical shifts were calibrated relative to the HOD chemical shift (4.7 ppm).

#### 4.6.3. Determination of the Degree of Deacetylation (DDA)

The degree of deacetylation of chitosan samples was measured by the titration method using a Hanna Hi 8733 (Hanna Instruments, Cluj-Napoca, Romania) conductometer. At first, 0.1 g of chitosan was weighed and dissolved in 5 mL 0.1 N HCl at room temperature. Distilled water (25 mL) was added, and the chitosan solution was then titrated against 0.1 M NaOH solution. A titration curve of pH values vs. NaOH titration volume was generated. DDA was calculated according to [54].

#### 4.6.4. High Performance Liquid Chromatography (HPLC)

HPLC was performed to determine the weight-average (M_w_) and number-average (M_n_) molecular weights and polydispersity indices (PDI = M_w_/M_n_) of chitosan samples by the method described by S.A. Lopatin et al. [55]. The HPLC apparatus S 2100 (Sykam, Ersesing, Germany) comprised a K-5004 degasser (Knauer, Berlin, Germany), Jet Stream + column thermostat (Knauer, Berlin, Germany), RI Detector K-2301 reflectometric detector (Knauer, Berlin, Germany), and PSS NOVEMA Max analytical 1000 A column (PSS, Mainz, Germany). Pullulans (M_w_ = 342, 1260, 6600, 9900, 23,000, 48,800, 113,000, 200,000, 348,000, and 805,000 Da) (PSS, Mainz, Germany) were used as the calibration standards.

### 4.7. Microbiological Assays

#### 4.7.1. Evaluation of the Antibacterial Activity

The antibacterial activity was studied according to the method described by Shagdarova et al. [56]. *Escherichia coli* ATCC 25922 and *Staphylococcus epidermidis* 33 GISK strains, deposited in the State Collection of Pathogenic Microorganisms of the Federal State Budgetary Institution “Scientific Centre for Expert Evaluation of Medicinal Products” of the Ministry of Health of the Russian Federation, were used. Bacterial cultures were stored in LA medium at 4 °C. To prepare the inoculum, a single bacterial colony was transferred to 20 mL of LB medium and incubated at 37 °C in a shaker at 150 rpm for 18 h. The substrates used for the evaluation of the antibacterial activity are presented in Table 4.

Minimum inhibitory concentrations of chitosan samples were determined by serial dilutions in a liquid medium. The experiment was carried out in a 96-well rounded-bottom plate. The chitosan samples were dissolved in 0.5% acetic acid at a concentration of 1 mg/mL. To determine the concentration dependence, a series of two-fold dilutions were made; the concentration range was 3.9–500 µg chitosan/mL of medium. Hence, the microbial load was 10^5^ cells/mL. Sterile culture medium (control of the purity of the medium), a suspension of test cultures in liquid culture medium without chitosan (control of growth), and 0.5% acetic acid (solvent for chitosan) were the experimental controls. Plates were incubated at 36 °C for 24 h. The results were evaluated visually, comparing the growth of the culture in the presence of a drug with a “negative control” containing the inoculum. The minimum concentration of the substance that completely suppressed the growth of the culture was defined as MIC.

#### 4.7.2. Evaluation of the Antifungal Activity

The experiment was carried out according to the method described in [57]. *Fusarium oxysporum* and *Botrytis cinerea* VKM F-1182 fungi were used in the experiments to determine the metabolic activity. Fungal strains were stored on potato dextrose agar (PDA) medium. The studied samples of chitosans were the same as in Section 4.7.1.

The MA of fungi upon the addition of chitosan samples to the nutrient medium was investigated in a liquid PDA medium at pH 5.5–5.7 using a modified tetrazolium method (MTT assay). The experiment was carried out in a 96-well flat-bottomed plate. To obtain conidia, 10 mL of liquid nutrient medium was added to a test tube with seven-day cultures on PDA, and a suspension containing conidia and fungal mycelium was prepared. The suspension was filtered through sterile cotton to remove the residual mycelium. Chitosan samples were dissolved in 0.5% acetic acid at a concentration of 15 mg/mL. To determine the concentration dependence, a series of two-fold dilutions was made; the concentration range was 0.234–1.875 mg chitosan/mL of medium. A suspension of conidia in a nutrient medium was added, while the concentration of conidia in the well was 2.5 × 10^4^ conidia/mL. The cultures were grown at 25 °C for 24 h. Then, 10 μL of iodonitrotetrazolium chloride solution (10 mg/mL) dissolved in 0.1 M phosphate buffer (pH 7.4) in the presence of 4 mg/mL 1-methoxy-phenazine-methosulfate was added to each well and incubated at 37 °C for 4 h. The supernatant liquid was removed, after which the formazan crystals were dissolved in 150 μL of DMSO at 37 °C with stirring (100 rpm) for 16 h. The optical density was then measured at 530 nm on a plate spectrophotometer (Thermo Scientific, Finland). The MA was calculated as follows:(1)MA=ODtestODcontrol
where *OD_test_* and *OD_control_* are the arithmetic mean of the optical density readings in the test and control samples, respectively. It is considered that a substance inhibits the metabolic activity of phytopathogens if MA is less than 50%.

## 5. Conclusions

Chitosan possesses a wide variety of biological activities, which explain the potential of this biopolymer in food, pharmaceutical, cosmetic, and agricultural industries. However, the properties are strongly influenced by the chemical composition of the molecules, especially their size, degree of deacetylation, and polydispersity. Therefore, controlled depolymerisation of well-characterised chitosans is of great importance for a deeper understanding of their bioactivity and thus their future applications.

To the best of our knowledge, this is the first study that investigates the influence of the chitosan’s source and enzymes used for depolymerisation on antibacterial and antifungal activities of the obtained low molecular weight chitosans. It was shown that the hydrolysates of insect and crab chitosans obtained under the same conditions using *T. harzianum* chitinase differ in M_w_ by almost 1.5 times. The studied chitosans differed in their degree of deacetylation, which can probably explain the differences in their depolymerisation behaviour. The enhanced antimicrobial activity of chitosan extracted from *H. illucens* larvae, especially in terms of antifungal activity against *F. oxysporum*, prove the significance of the chitosan origin. Overall, it was shown that the source of chitosan and the applied enzyme preparation impact the properties of biopolymers and are, therefore, a promising area for future research.

## Figures and Tables

**Figure 1 molecules-27-00577-f001:**
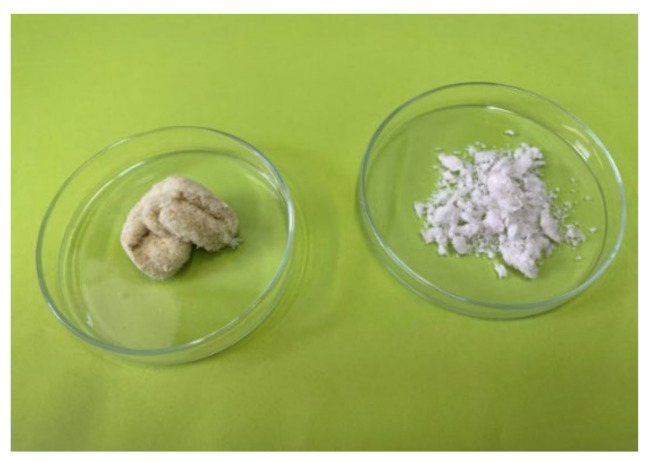
Cuticulin (left) and reprecipitated chitosan (right).

**Figure 2 molecules-27-00577-f002:**
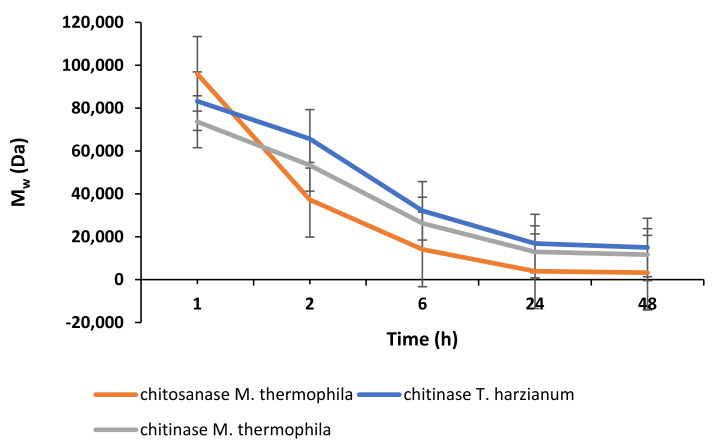
Kinetics of the changes in molecular weight of chitosan from *H. illucens* larvae during enzymatic hydrolysis.

**Figure 3 molecules-27-00577-f003:**
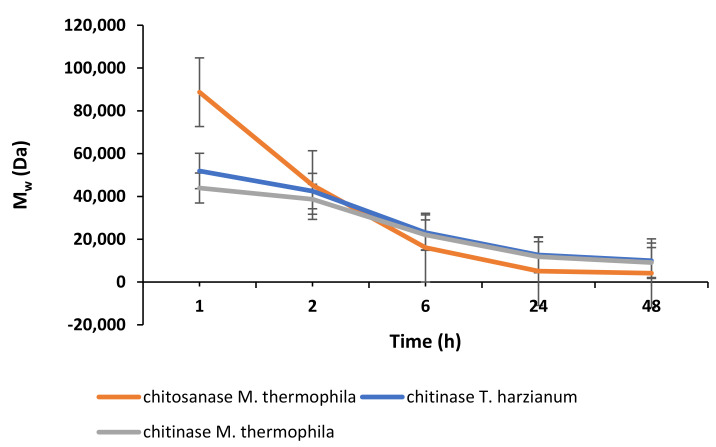
Kinetics of changes in the molecular weight of crab chitosan during enzymatic hydrolysis.

**Figure 4 molecules-27-00577-f004:**
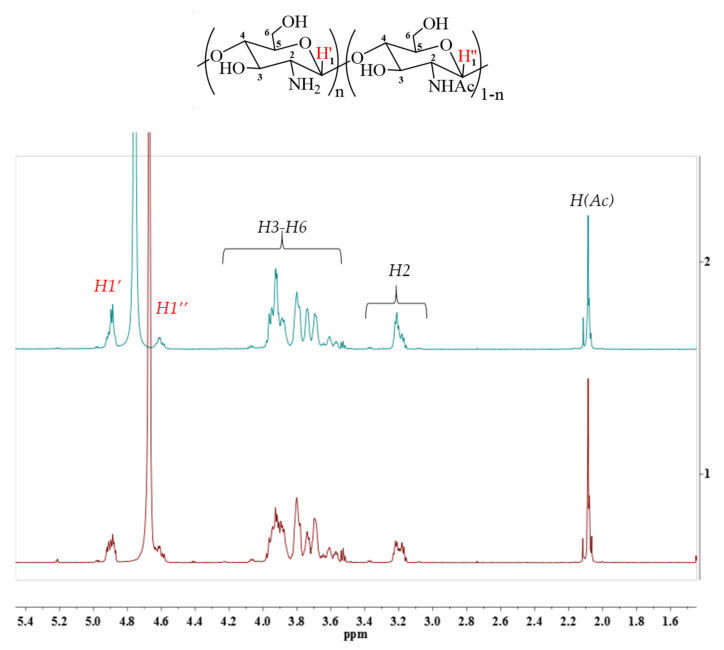
^1^H NMR spectra of depolymerised chitosans obtained from *H. illucens* (6.86 kDa) (**2**) and crab (6.78 kDa) (**1**).

**Figure 5 molecules-27-00577-f005:**
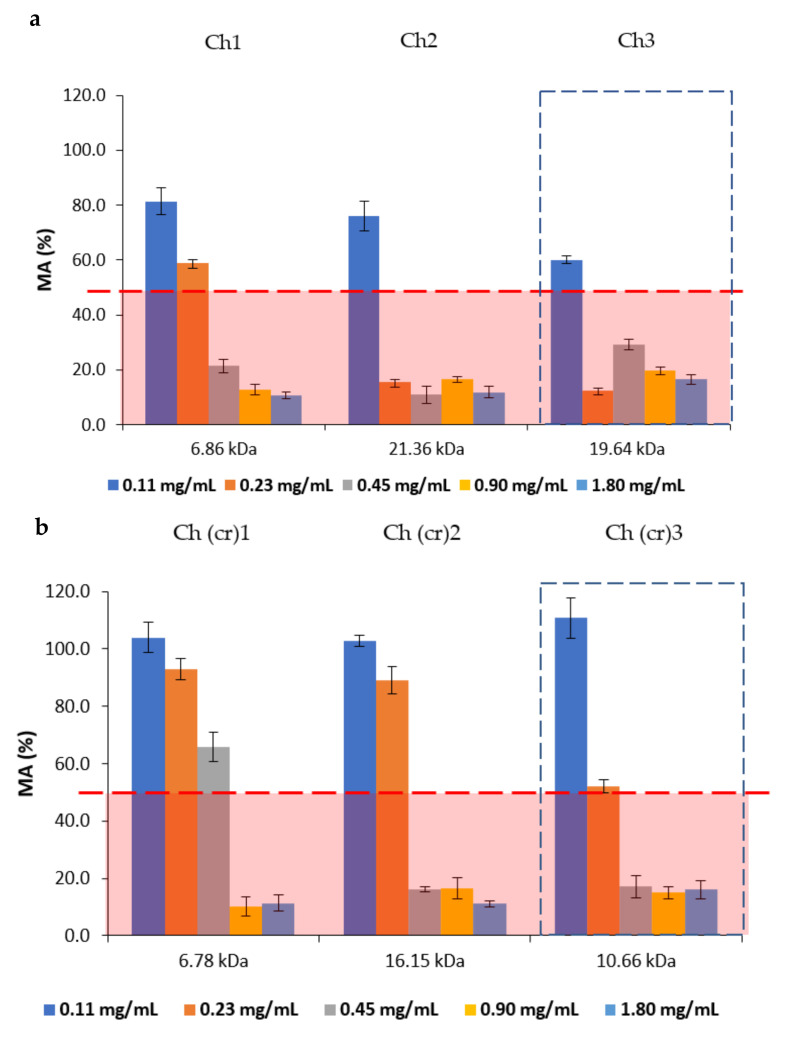
Influence of molecular weight and concentration of chitosan samples obtained from *H. illucens* larvae (**a**) and crab (**b**) on the metabolic activity (MA) of *F. oxysporum*.

**Table 1 molecules-27-00577-t001:** Elemental analyses data for depolymerised chitosans obtained from crab (6.78 kDa) and *H. illucens* (6.86 kDa).

Source	%C	%H	%N
Crab	39.48	6.91	6.67
*H. illucens*	41.28	6.87	7.09

**Table 2 molecules-27-00577-t002:** The antibacterial activities of crab and insect chitosan samples against *S. epidermidis*. Ch—insect chitosan; Ch (cr)—crab chitosan.

Sample	Ch1	Ch (cr)1	Ch2	Ch (cr)2	Ch3	Ch (cr)3
Mw, kDa	6.86	6.78	21.36	16.15	19.64	10.66
PDI	1.67	1.54	2.14	1.95	1.96	1.71
MIC, µg/mL	62.5	125	62.5	62.5	62.5	62.5

**Table 3 molecules-27-00577-t003:** The activity of the enzyme preparations towards chitosan (1000 kDa).

Enzyme Preparation Number	Recombinant Enzyme	Activity by Chitosan (1000 kDa), U/g *
# 3-432.5	chitosanase Chi 402 from *M. thermophila* (M_w_ 44 kDa)	600
# 3-458.1	chitinase Chi 418 from *T. harzianum* (M_w_ 42 kDa)	2000
# 3-544.H	chitinase Chi 403 from *M. thermophila* (M_w_ 43 kDa)	1500

* by K_3_Fe(CN)_6_ method [53].

**Table 4 molecules-27-00577-t004:** Substrates used in the experiment.

Sample	M_W_, kDa	PDI	Enzyme
Ch1	6.86	1.67	# 3-432.5 Chi 402—chitosanase *M. thermophila*
Ch (cr)1	6.78	1.54	
Ch2	21.36	2.14	
Ch (cr)2	16.15	1.95	
Ch3	19.64	1.96	# 3-544.H Chi 403—chitinase *M. thermophila*
Ch (cr)3	10.66	1.71	

## Data Availability

Not applicable.

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
