# Peer review of "Evaluation of Antibacterial and Antifungal Properties of Low Molecular Weight Chitosan Extracted from Hermetia illucens Relative to Crab Chitosan"

_molecules, 2022, doi:10.3390/molecules27020577_

Round 1

Reviewer 1 Report

The manuscript titled "Evaluation of antibacterial and antifungal properties of low molecular weight chitosan extracted from Hermetia illucens relative to crab chitosan" is interesting. After all, antimicrobial properties of chitosan is one of the important application areas of this important polysaccharide . I recommend publication of this manuscript after author take care of many typos all over the places in the manuscript.

Author Response

Response to Reviewer 1 Comments

Thank you for giving us the opportunity to submit a revised draft of the manuscript “Evaluation of Antibacterial and Antifungal Properties of Low Molecular Weight Chitosan Extracted from Hermetia illucens Relative to Crab Chitosan” for publication in Molecules. We appreciate the time and effort that you dedicated to providing feedback on our manuscript and are grateful for the valuable comments on our paper. We have corrected the typos (marked up in text), such as thermostating, dialysed, and improved the writing style.

Yours sincerely,

on behalf of all co-authors,

Adelya Khayrova

10-01-2022

Reviewer 2 Report

The paper “Evaluation of Antibacterial and Antifungal Properties of Low Molecular Weight Chitosan Extracted from Hermetia illucens Relative to Crab Chitosan” by Adelya Khayrova et al, investigates the depolymerization  of chitosan from different origins by three different enzymes and the antimicrobial activity of the resulting low molecular weight oligomers. The compounds investigated and compared in this research are Chitosan from Hermetia illucens (Black soldier fly larvae) and commercial crab chitosan and the microorganisms tested are S. Epidermis, E. Coli for antibacterial activity and B. cinerea and F. Oxysporum for antifungal activity.

The research shown herein provides experimental data for non-conventional chitin and chitosan sources such as fly larvae. The physicochemical properties characterized by the authors are important aspects of chitosan in terms of its antimicrobial activity and the methodology used is in general appropriate.

The microbiological section of the paper shows several problems in terms of the results presentation and discussion:

  1. Table 2. The way of presenting the data is confusing; PDI is shown in the table but never discussed or related to the MICs.
  2. Figure 5. Sorting the results by increasing molecular weight would be clearer. 
  3. Line 121. The authors must provide the appropriate citations that indicates that metabolic activity and IC50 concepts can be related directly. IC50 is commonly related to fungal growth (mycelial and sporulation) whereas metabolic activity assessed by MTT is related to mitochondrial dehydrogenases.
  4. The data at a concentration of 225 mg/mL for the 19.64 kDa chitosan sample obtained from H. 128 illucens larvae do not follow a trend. Is there an explanation to this?
  5. Line 145. The authors don’t show any explicit number on the molecular weight results section, so it is difficult for the reader to estimate that one chitinase is 1.5 times better to hydrolyze chitosan.
  6. Line 147. The results show that different chitinases have different effect on chitosan hydrolysis, but the source of the chitosan doesn’t show any impact on the final molecular weight. Again, if explicit molecular weight data is not shown, it is difficult to compare these results.
  7. It is recommended to deepen the discussion on the results that indicate that chitosan is more effective on gram positive bacteria than gram negative bacteria, addressing the electrostatic effects.
  8. Line 180.  Isn't low molecular weight chitosan able to penetrate the outer membrane of gram-negative bacteria?
  9. Radial growth experiments to test the antifungal effect of chitosan instead of MTT assay are highly recommended.

Author Response

Response to Reviewer 2 Comments

Thank you for giving us the opportunity to submit a revised draft of the manuscript “Evaluation of Antibacterial and Antifungal Properties of Low Molecular Weight Chitosan Extracted from Hermetia illucens Relative to Crab Chitosan” for publication in Molecules. We appreciate the time and effort that you dedicated to providing feedback on our manuscript and are grateful for the valuable comments on our paper. We have been able to incorporate changes to reflect most of your suggestions. Those changes are highlighted within the manuscript.

Please see below, in red, for a point-by-point response to the comments and concerns.

Point 1: Table 2. The way of presenting the data is confusing; PDI is shown in the table but never discussed or related to the MICs.

Response 1: Thank you for pointing this out. Polydispersity index has been discussed further as follows:

The studied chitosan samples had low polydispersity index (PDI=1.5-2). Differences in PDI values of studied chitosan pairs did not exceed 10-15% corresponding to the correctness in comparison of samples.

Point 2: Figure 5. Sorting the results by increasing molecular weight would be clearer.

Response 2: We agree with this and have incorporated your suggestion into the manuscript by labelling the chitosan samples in Fig. 5 (e.g. Ch1, Ch (cr)1 etc).

Point 3: Line 121. The authors must provide the appropriate citations that indicates that metabolic activity and IC50 concepts can be related directly. IC50 is commonly related to fungal growth (mycelial and sporulation) whereas metabolic activity assessed by MTT is related to mitochondrial dehydrogenases.

Response 3: Thank you for this suggestion. We have added an appropriate citation (Y. Park et al., 2008) to the manuscript.

Point 4: The data at a concentration of 225 mg/mL for the 19.64 kDa chitosan sample obtained from H. 128 illucens larvae do not follow a trend. Is there an explanation to this?

Response 4: Thank you for pointing this out. We are unable to explain this discrepancy, however, the experiment was repeated three times.

Point 5: Line 145. The authors don’t show any explicit number on the molecular weight results section, so it is difficult for the reader to estimate that one chitinase is 1.5 times better to hydrolyze chitosan.

Response 5: Agree. We have, accordingly, added some numbers to emphasise this point:

p.6. Nevertheless, the hydrolysates of chitosans obtained under the same conditions using chitinase T. harzianum differ in Mw by almost 1.5 times (e.g. 14995 Da versus 9959 Da).

Point 6: Line 147. The results show that different chitinases have different effect on chitosan hydrolysis, but the source of the chitosan doesn’t show any impact on the final molecular weight. Again, if explicit molecular weight data is not shown, it is difficult to compare these results.

Response 6: Thank you for this comment. The source of the chitosan shows an impact on the final molecular weight. As stated earlier (p. 3), based on the kinetic curves (Fig. 2 and 3) of the accumulation of the conversion products of chitosans, hydrolysis time to obtain chitosans with specified molecular weights was estimated.

Point 7: It is recommended to deepen the discussion on the results that indicate that chitosan is more effective on gram positive bacteria than gram negative bacteria, addressing the electrostatic effects.

Response 7: Thank you for this suggestion. We have added an appropriate citation (J. Li and S. Zhuang, 2020) to the manuscript (p. 6).

This could be explained by the contrasting mechanisms of interaction of chitosan with gram-positive and gram-negative bacterial cells due to their structural differences as well as the electrostatic effects.

Point 8: Line 180.  Isn't low molecular weight chitosan able to penetrate the outer membrane of gram-negative bacteria?

Response 8: According to our data, low molecular weight chitosan is unable to penetrate the outer membrane of gram-negative bacteria. Molecular weight could not be lowered by using the mentioned enzyme preparations.

Point 9: Radial growth experiments to test the antifungal effect of chitosan instead of MTT assay are highly recommended.

Response 9: Thank you for this comment. Radial growth method is used for initial screening and is less accurate comparing to spectrophotometric method (MTT assay). (S. S. Terekhov et al., 2018 https://actanaturae.ru/2075-8251/article/view/10317/121)

In addition to the above comments, spelling and grammatical errors have been corrected.

Yours sincerely,

on behalf of all co-authors,

Adelya Khayrova

10-01-2022

Round 2

Reviewer 2 Report

The authors have made the appropriate revisions to their manuscript. In the last reply, the reference discussed is for bacterial growth, not fungal growth. However, this is a minor point and the manuscritp can be published in the current form.

This manuscript is a resubmission of an earlier submission. The following is a list of the peer review reports and author responses from that submission.